# A Prior of a Googol Gaussians: a Tensor Ring Induced Prior for Generative Models

**Maksim Kuznetsov**[1,*]    **Daniil Polykovskiy**[1,*]    **Dmitry Vetrov**[2]    **Alexander Zhebrak**[1]

[1]Insilico Medicine   [2]National Research University Higher School of Economics
{kuznetsov,daniil,zhebrak}@insilico.com   vetrovd@yandex.ru

## Abstract

Generative models produce realistic objects in many domains, including text, image, video, and audio synthesis. Most popular models—Generative Adversarial Networks (GANs) and Variational Autoencoders (VAEs)—usually employ a standard Gaussian distribution as a prior. Previous works show that the richer family of prior distributions may help to avoid the mode collapse problem in GANs and to improve the evidence lower bound in VAEs. We propose a new family of prior distributions—Tensor Ring Induced Prior (TRIP)—that packs an exponential number of Gaussians into a high-dimensional lattice with a relatively small number of parameters. We show that these priors improve Fréchet Inception Distance for GANs and Evidence Lower Bound for VAEs. We also study generative models with TRIP in the conditional generation setup with missing conditions. Altogether, we propose a novel plug-and-play framework for generative models that can be utilized in any GAN and VAE-like architectures.

## 1   Introduction

Modern generative models are widely applied to the generation of realistic and diverse images, text, and audio files [1–5]. Generative Adversarial Networks (GAN) [6], Variational Autoencoders (VAE) [7], and their variations are the most commonly used neural generative models. Both architectures learn a mapping from some prior distribution $p(z)$—usually a standard Gaussian—to the data distribution $p(x)$. Previous works showed that richer prior distributions might improve the generative models—reduce mode collapse for GANs [8, 9] and obtain a tighter Evidence Lower Bound (ELBO) for VAEs [10].

If the prior $p(z)$ lies in a parametric family, we can learn the most suitable distribution for it during training. In this work, we investigate Gaussian Mixture Models as prior distributions with an exponential number of Gaussians in nodes of a multidimensional lattice. In our experiments, we used a prior with more than a googol ($10^{100}$) Gaussians. To handle such complex distributions, we represent $p(z)$ using a Tensor Ring decomposition [11]—a method for approximating high-dimensional tensors with a relatively small number of parameters. We call this family of distributions a Tensor Ring Induced Prior (TRIP). For this distribution, we can compute marginal and conditional probabilities and sample from them efficiently.

We also extend TRIP to conditional generation, where a generative model $p(x \mid y)$ produces new objects $x$ with specified attributes $y$. With TRIP, we can produce new objects conditioned only on a subset of attributes, leaving some labels unspecified during both training and inference.

Our main contributions are summarized as follows:

---

[*]equal contribution

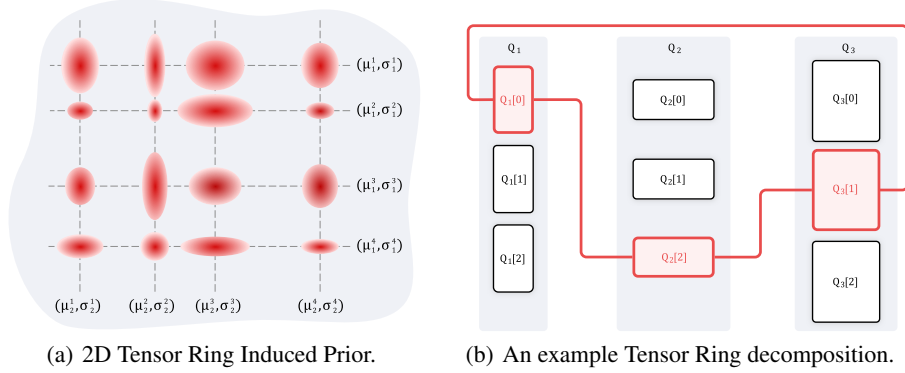

(a) 2D Tensor Ring Induced Prior.　　　　(b) An example Tensor Ring decomposition.

Figure 1: **(a)** The TRIP distribution is a multidimensional Gaussian Mixture Model with an exponentially large number of modes located on the lattice nodes. **(b)** To compute the value $\widehat{P}[0, 2, 1]$, one should multiply the highlighted matrices and compute the trace $\widehat{P}[0, 2, 1] = \mathrm{Tr}(Q_1[0] \cdot Q_2[2] \cdot Q_3[1])$.

- We introduce a family of distributions that we call a Tensor Ring Induced Prior (TRIP) and use it as a prior for generative models—VAE, GAN, and its variations.
- We investigate an application of TRIP to conditional generation and show that this prior improves quality on sparsely labeled datasets.
- We evaluate TRIP models on the generation of CelebA faces for both conditional and unconditional setups. For GANs, we show improvement in Fréchet Inception Distance (FID) and improved ELBO for VAEs. For the conditional generation, we show lower rates of condition violation compared to standard conditional models.

## 2　Tensor Ring Induced Prior

In this section, we introduce a Tensor Ring-induced distribution for both discrete and continuous variables. We also define a Tensor Ring Induced Prior (TRIP) family of distributions.

### 2.1　Tensor Ring decomposition

Tensor Ring decomposition [11] represents large high-dimensional tensors (such as discrete distributions) with a relatively small number of parameters. Consider a joint distribution $p(r_1, r_2, \ldots r_d)$ of $d$ discrete random variables $r_k$ taking values from $\{0, 1, \ldots N_k - 1\}$. We write these probabilities as elements of a $d$-dimensional tensor $P[r_1, r_2, \ldots r_d] = p(r_1, r_2, \ldots r_d)$. For the brevity of notation, we use $r_{1:d}$ for $(r_1, \ldots, r_d)$. The number of elements in this tensor grows exponentially with the number of dimensions $d$, and for only 50 binary variables the tensor contains $2^{50} \approx 10^{15}$ real numbers. Tensor Ring decomposition reduces the number of parameters by approximating tensor $P$ with low-rank non-negative tensors *cores* $Q_k \in \mathbb{R}_+^{N_k \times m_k \times m_{k+1}}$, where $m_1, \ldots, m_{d+1}$ are core sizes, and $m_{d+1} = m_1$:

$$p(r_{1:d}) \propto \widehat{P}[r_{1:d}] = \mathrm{Tr}\Big(\prod_{j=1}^{d} Q_j[r_j]\Big) \tag{1}$$

To compute $P[r_{1:d}]$, for each random variable $r_k$, we slice a tensor $Q_k$ along the first dimension and obtain a matrix $Q_k[r_k] \in \mathbb{R}_+^{m_k \times m_{k+1}}$. We multiply these matrices for all random variables and compute the trace of the resulting matrix to get a scalar (see Figure 1(b) for an example). In Tensor Ring decomposition, the number of parameters grows linearly with the number of dimensions. With larger core sizes $m_k$, Tensor Ring decomposition can approximate more complex distributions. Note that the order of the variables matters: Tensor Ring decomposition better captures dependencies between closer variables than between the distant ones.

With Tensor Ring decomposition, we can compute marginal distributions without computing the whole tensor $\widehat{P}[r_{1:d}]$. To marginalize out the random variable $r_k$, we replace cores $Q_k$ in Eq 1 with

matrix $\widetilde{Q}_k = \sum_{r_k=0}^{N_k-1} Q_k[r_k]$:

$$p(r_{1:k-1}, r_{k+1:d}) \propto \widehat{P}[r_{1:k-1}, r_{k+1:d}] = \mathrm{Tr}\left( \prod_{j=1}^{k-1} Q_j[r_j] \cdot \widetilde{Q}_k \cdot \prod_{j=k+1}^{d} Q_j[r_j] \right) \tag{2}$$

In Supplementary Materials, we show an Algorithm for computing marginal distributions. We can also compute conditionals as a ratio between the joint and marginal probabilities $p(A \mid B) = p(A, B)/p(B)$; we sample from conditional or marginal distributions using the chain rule.

## 2.2 Continuous Distributions parameterized with Tensor Ring Decomposition

In this section, we apply the Tensor Ring decomposition to continuous distributions over vectors $z = [z_1, \ldots, z_d]$. In our Learnable Prior model, we assume that each component of $z_k$ is a Gaussian Mixture Model with $N_k$ fully factorized components. The joint distribution $p(z)$ is a multidimensional Gaussian Mixture Model with modes placed in the nodes of a multidimensional lattice (Figure 1(a)). The latent discrete variables $s_1, \ldots, s_d$ indicate the index of mixture component for each dimension ($s_k$ corresponds to the $k$-th dimension of the latent code $z_k$):

$$p(z_{1:d}) = \sum_{s_{1:d}} p(s_{1:d}) p(z_{1:d} \mid s_{1:d}) \propto \sum_{s_{1:d}} \widehat{P}[s_{1:d}] \prod_{j=1}^{d} \mathcal{N}(z_j \mid \mu_j^{s_j}, \sigma_j^{s_j}) \tag{3}$$

Here, $p(s)$ is a discrete distribution of prior probabilities of mixture components, which we store as a tensor $\widehat{P}[s]$ in a Tensor Ring decomposition. Note that $p(s)$ is not a factorized distribution, and the learnable prior $p(z)$ may learn complex weightings of the mixture components. We call the family of distributions parameterized in this form a Tensor Ring Induced Prior (TRIP) and denote its learnable parameters (cores, means, and standard deviations) as $\psi$:

$$\psi = \left\{ Q_1, \ldots, Q_d, \mu_1^0, \ldots, \mu_d^{N_d-1}, \sigma_1^0, \ldots, \sigma_d^{N_d-1} \right\}. \tag{4}$$

To highlight that the prior distribution is learnable, we further write it as $p_\psi(z)$. As we show later, we can optimize $\psi$ directly using gradient descent for VAE models and REINFORCE [12] for GANs.

An important property of the proposed TRIP family is that we can derive its one-dimensional conditional distributions in a closed form. For example, to sample using a chain rule, we need distributions $p_\psi(z_k \mid z_{1:k-1})$:

$$
\begin{aligned}
p_\psi(z_k \mid z_{1:k-1}) &= \sum_{s_k=0}^{N_k-1} p_\psi(s_k \mid z_{1:k-1}) p_\psi(z_k \mid s_k, z_{1:k-1}) \\
&= \sum_{s_k=0}^{N_k-1} p_\psi(s_k \mid z_{1:k-1}) p_\psi(z_k \mid s_k) = \sum_{s_k=0}^{N_k-1} p_\psi(s_k \mid z_{1:k-1}) \mathcal{N}(z_k \mid \mu_k^{s_k}, \sigma_k^{s_k})
\end{aligned}
\tag{5}
$$

From Equation 5 we notice that one-dimensional conditional distributions are Gaussian Mixture Models with the same means and variances as priors, but with different weights $p_\psi(s_k \mid z_{1:k-1})$ (see Supplementary Materials).

Computations for marginal probabilities in the general case are shown in Algorithm 1; conditional probabilities can be computed as a ratio between the joint and marginal probabilities. Note that we compute a normalizing constant on-the-fly.

## 3 Generative Models With Tensor Ring Induced Prior

In this section, we describe how popular generative models—Variational Autoencoders (VAEs) and Generative Adversarial Networks (GANs)—can benefit from using Tensor Ring Induced Prior.

### 3.1 Variational Autoencoder

Variational Autoencoder (VAE) [7, 13] is an autoencoder-based generative model that maps data points $x$ onto a latent space with a probabilistic encoder $q_\phi(z \mid x)$ and reconstructs objects with a probabilistic decoder $p_\theta(x \mid z)$. We used a Gaussian encoder with the reparameterization trick:

$$q_\phi(z \mid x) = \mathcal{N}(z \mid \mu_\phi(x), \sigma_\phi(x)) = \mathcal{N}(\epsilon \mid 0, I) \cdot \sigma_\phi(x) + \mu_\phi(x). \tag{6}$$

**Algorithm 1** Calculation of marginal probabilities in TRIP
___
**Input:** A set $M$ of variable indices for which we compute the probability, and values of these latent codes $z_i$ for $i \in M$
**Output:** Joint probability $\log p(z_M)$, where $z_M = \{z_i \, \forall i \in M\}$
Initialize $Q_{\text{buff}} = I \in \mathbb{R}^{m_1 \times m_1}$, $Q_{\text{norm}} = I \in \mathbb{R}^{m_1 \times m_1}$
**for** $j = 1$ **to** $d$ **do**
    **if** $j$ is marginalized out $(j \notin M)$ **then**
        $Q_{\text{buff}} = Q_{\text{buff}} \cdot \left( \sum_{k=0}^{N_j-1} Q_j[k] \right)$
    **else**
        $Q_{\text{buff}} = Q_{\text{buff}} \cdot \left( \sum_{k=0}^{N_j-1} Q_j[k] \cdot \mathcal{N}\left( z_k \mid \mu_j^{s_j}, \sigma_j^{s_j} \right) \right)$
    **end if**
    $Q_{\text{norm}} = Q_{\text{norm}} \cdot \left( \sum_{k=0}^{N_j-1} Q_j[k] \right)$
**end for**
$\log p(z_M) = \log \text{Tr}\left(Q_{\text{buff}}\right) - \log \text{Tr}\left(Q_{\text{norm}}\right)$
___

The most common choice for a prior distribution $p_\psi(z)$ in the latent space is a standard Gaussian distribution $\mathcal{N}(0, I)$. VAEs are trained by maximizing the lower bound of the log marginal likelihood $\log p(x)$, also known as the Evidence Lower Bound (ELBO):

$$\mathcal{L}(\theta, \phi, \psi) = \mathbb{E}_{q_\phi(z|x)} \log p_\theta(x \mid z) - \mathcal{KL}\big(q_\phi(z \mid x) \, || \, p_\psi(z)\big), \tag{7}$$

where $\mathcal{KL}$ is a Kullback-Leibler divergence. We get an unbiased estimate of $\mathcal{L}(\theta, \phi, \psi)$ by sampling $\epsilon_i \sim \mathcal{N}(0, I)$ and computing a Monte Carlo estimate

$$\mathcal{L}(\theta, \phi, \psi) \approx \frac{1}{l} \sum_{i=1}^{l} \log \left( \frac{p_\theta(x \mid z_i) p_\psi(z_i)}{q_\phi(z_i \mid x)} \right), \quad z_i = \epsilon_i \cdot \sigma_\phi(x) + \mu_\phi(x) \tag{8}$$

When $p_\psi(z)$ is a standard Gaussian, the $\mathcal{KL}$ term can be computed analytically, reducing the estimation variance.

For VAEs, flexible priors give tighter evidence lower bound [10, 14] and can help with a problem of the decoder ignoring the latent codes [14, 15]. In this work, we parameterize the learnable prior $p_\psi(z)$ as a Tensor Ring Induced Prior model and train its parameters $\psi$ jointly with encoder and decoder (Figure 2). We call this model a Variational Autoencoder with Tensor Ring Induced Prior (VAE-TRIP). We initialize the means and the variances by fitting 1D Gaussian Mixture models for each component using samples from the latent codes and initialize cores with a Gaussian noise. We then re-initialize means, variances and cores after the first epoch, and repeat such procedure every 5 epochs.

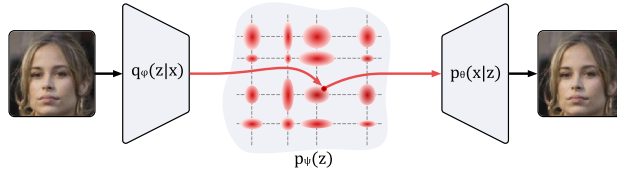

Figure 2: A Variational Autoencoder with a Tensor Ring Induced Prior (VAE-TRIP).

## 3.2 Generative Adversarial Networks

Generative Adversarial Networks (GANs) [6] consist of two networks: a generator $G(z)$ and a discriminator $D(x)$. The discriminator is trying to distinguish real objects from objects produced by a generator. The generator, on the other hand, is trying to produce objects that the discriminator considers real. The optimization setup for all models from the GAN family is a min-max problem. For the standard GAN, the learning procedure alternates between optimizing the generator and the

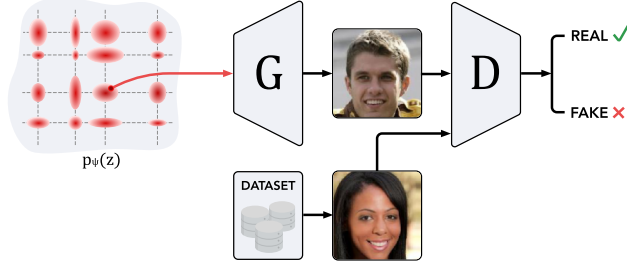

Figure 3: A Generative Adversarial Network with a Tensor Ring Induced Prior (GAN-TRIP).

discriminator networks with a gradient descent/ascent:

$$\min_{G,\psi} \max_D \mathcal{L}_{GAN} = \mathbb{E}_{x \sim p(x)} \log D(x) + \mathbb{E}_{z \sim p_\psi(z)} \log \left(1 - D\big(G(z)\big)\right) \tag{9}$$

Similar to VAE, the prior distribution $p_\psi(z)$ is usually a standard Gaussian, although Gaussian Mixture Models were also previously studied [16]. In this work, we use a TRIP family of distributions to parameterize a multimodal prior of GANs (Figure 3). We expect that having multiple modes as the prior improves the overall quality of generation and helps to avoid anomalies during sampling, such as partially present eyeglasses.

During training, we sample multiple latent codes from the prior $p_\psi(z)$ and use REINFORCE [12] to propagate the gradient through the parameters $\psi$. We reduce the variance by using average discriminator output as a baseline:

$$\nabla_\psi \mathcal{L}_{GAN} \approx \frac{1}{l} \sum_{i=1}^{l} \nabla_\psi \log p_\psi(z_i) \left[ d_i - \frac{1}{l} \sum_{j=1}^{l} d_j \right], \tag{10}$$

where $d_i = \log \left(1 - D\big(G(z)\big)\right)$ is the discriminator's output and $z_i$ are samples from the prior $p_\psi(z)$. We call this model a Generative Adversarial Network with Tensor Ring Induced Prior (GAN-TRIP). We initialize means uniformly in a range $[-1, 1]$ and standard deviations as $1/N_k$.

## 4    Conditional Generation

In conditional generation problem, data objects $x$ (for example, face images) are coupled with properties $y$ describing the objects (for example, sex and hair color). The goal of this model is to learn a distribution $p(x \mid y)$ that produces objects with specified attributes. Some of the attributes $y$ for a given $x$ may be unknown ($y_{\text{un}}$), and the model should learn solely from observed attributes ($y_{\text{ob}}$): $p(x \mid y_{\text{ob}})$.

For VAE-TRIP, we train a joint model $p_\psi(z, y)$ on all attributes $y$ and latent codes $z$ parameterized with a Tensor Ring. For discrete conditions, the joint distribution is:

$$p(z, y) = \sum_{s_{1:d}} \widetilde{P}[s_{1:d}, y] \prod_{j=1}^{d} \mathcal{N}(z_d \mid \mu_d^{s_d}, \sigma_d^{s_d}), \tag{11}$$

where tensor $\widetilde{P}[s_{1:d}, y]$ is represented in a Tensor Ring decomposition. In this work, we focus on discrete attributes, although we can extend the model to continuous attributes with Gaussian Mixture Models as we did for the latent codes.

With the proposed parameterization, we can marginalize out missing attributes and compute conditional probabilities. We can efficiently compute both probabilities similar to Algorithm 1.

For conditional VAE model, the lower bound on $\log p(x, y_{\text{ob}})$ is:

$$\widetilde{\mathcal{L}}(\theta, \phi, \psi) = \mathbb{E}_{q_\phi(z \mid x, y_{\text{ob}})} \log p_\theta(x, y_{\text{ob}} \mid z) - \mathcal{KL}\big(q_\phi(z \mid x, y_{\text{ob}}) \,\|\, p_\psi(z)\big). \tag{12}$$

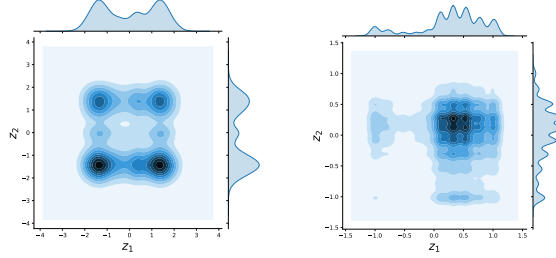

Figure 4: Visualization of the first two dimensions of the learned prior $p_\psi(z_1, z_2)$. **Left:** VAE-TRIP, **Right:** WGAN-GP-TRIP.

We simplify the lower bound by making two restrictions. First, we assume that the conditions $y$ are fully defined by the object $x$, which implies $q_\phi(z \mid x, y_{\text{ob}}) = q_\phi(z \mid x)$. For example, an image with a person wearing a hat defines the presence of a hat. The second restriction is that we can reconstruct an object directly from its latent code: $p_\theta(x \mid z, y_{\text{ob}}) = p_\theta(x \mid z)$. This restriction also gives:

$$p_\theta(x, y_{\text{ob}} \mid z) = p_\theta(x \mid z, y_{\text{ob}})p_\psi(y_{\text{ob}} \mid z) = p_\theta(x \mid z)p_\psi(y_{\text{ob}} \mid z). \tag{13}$$

The resulting Evidence Lower Bound is

$$\widetilde{\mathcal{L}}(\theta, \phi, \psi) = \mathbb{E}_{q_\phi(z|x)}\big[\log p_\theta(x \mid z) + \log p_\psi(y_{\text{ob}} \mid z)\big] - \mathcal{KL}\big(q_\phi(z \mid x) \mid\mid p_\psi(z)\big). \tag{14}$$

In the proposed model, an autoencoder learns to map objects onto a latent manifolds, while TRIP prior $\log p_\psi(z \mid y_{\text{ob}})$ finds areas on the manifold corresponding to objects with the specified attributes.

The quality of the model depends on the order of the latent codes and the conditions in $p_\psi(z, y)$, since the Tensor Ring poorly captures dependence between variables that are far apart. In our experiments, we found that randomly permuting latent codes and conditions gives good results.

We can train the proposed model on partially labeled datasets and use it to draw conditional samples with partially specified constraints. For example, we can ask the model to generate images of men in hats, not specifying hair color or the presence of glasses.

## 5  Related Work

The most common generative models are based on Generative Adversarial Networks [6] or Variational Autoencoders [7]. Both GAN and VAE models usually use continuous unimodal distributions (like a standard Gaussian) as a prior. A space of natural images, however, is multimodal: a person either wears glasses or not—there are no intermediate states. Although generative models are flexible enough to transform unimodal distributions to multimodal, they tend to ignore some modes (mode collapse) or produce images with artifacts (half-present glasses).

A few models with learnable prior distributions were proposed. Tomczak and Welling [10] used a Gaussian mixture model based on encoder proposals as a prior on the latent space of VAE. Chen et al. [14] and Rezende and Mohamed [17] applied normalizing flows [18–20] to transform a standard normal prior into a more complex latent distribution. [14, 15] applied auto-regressive models to learn better prior distribution over the latent variables. [21] proposed to update a prior distribution of a trained VAE to avoid samples that have low marginal posterior, but high prior probability.

Similar to Tensor Ring decomposition, a Tensor-Train decomposition [22] is used in machine learning and numerical methods to represent tensors with a small number of parameters. Tensor-Train was applied to the compression of fully connected [23], convolutional [24] and recurrent [25] layers. In our models, we can use a Tensor-Train decomposition instead of Tensor Ring, but it requires larger cores to achieve comparable results, as first and last dimensions are farther apart.

Most conditional models work with missing values by imputing them with a predictive model or setting them to a special value. With this approach, we cannot sample objects specifying conditions partially. VAE TELBO model [26] proposes to train a Product of Experts-based model, where the posterior on the latent codes is approximated as $p_\psi(z \mid y_{\text{ob}}) = \prod_{y_i \in y_{\text{ob}}} p_\psi(z \mid y_i)$, requiring to train

a separate posterior model for each condition. JMVAE model [27] contains three encoders that take both image and condition, only a condition, or only an image.

# 6 Experiments

We conducted experiments on CelebFaces Attributes Dataset (CelebA) [28] of approximately 400,000 photos with a random train-test split. For conditional generation, we selected 14 binary image attributes, including sex, hair color, presence mustache, and beard. We compared both GAN and VAE models with and without TRIP. We also compared our best model with known approaches on CIFAR-10 [29] dataset with a standard split. Model architecture and training details are provided in Supplementary Materials.

## 6.1 Generating Objects With VAE-TRIP and GAN-TRIP

Table 1: FID for GAN and VAE-based architectures trained on CelebA dataset, and ELBO for VAE. F = Fixed, L = Learnable. We also report ELBO for importance-weighted autoencoder with $k = 100$ points [30]

| METRIC | MODEL | $\mathcal{N}(0, I)$ | GMM | | TRIP | |
|---|---|---|---|---|---|---|
| | | | F | L | F | L |
| FID | VAE | 86.72 | 85.64 | 84.48 | 85.31 | **83.54** |
| | WGAN | 63.46 | 67.10 | 61.82 | 62.48 | **57.6** |
| | WGAN-GP | 54.71 | 57.82 | 62.10 | 63.06 | **52.86** |
| ELBO | VAE | -194.16 | -201.60 | -193.88 | -202.04 | **-193.32** |
| IWAE ELBO ($k = 100$) | | -185.09 | -191.99 | -184.73 | -190.09 | **-184.43** |

We evaluate GAN-based models with and without Tensor Ring Learnable Prior by measuring a Fréchet Inception Distance (FID). For the baseline models, we used Wasserstein GAN (WGAN) [31] and Wasserstein GAN with Gradient Penalty (WGAN-GP) [32] on CelebA dataset. We also compared learnable priors with fixed randomly initialized parameters $\psi$. The results in Table 1 (CelebA) and Table 2 (CIFAR-10) suggest that with a TRIP prior the quality improves compared to standard models and models with GMM priors. In some experiments, the GMM-based model performed worse than a standard Gaussian, since $\mathcal{KL}$ had to be estimated with Monte-Carlo sampling, resulting in higher gradient variance.

Table 2: FID for CIFAR-10 GAN-based models

| Model | FID |
|---|---|
| SN-GANs [33] | 21.7 |
| WGAN-GP + Two Time-Scale [34] | 24.8 |
| WGAN-GP [32] | 29.3 |
| WGAN-GP-TRIP (ours) | **16.72** |

## 6.2 Visualization of TRIP

In Figure 4, we visualize first two dimensions of the learned prior $p_\psi(z_1, z_2)$ in VAE-TRIP and WGAN-GP-TRIP models. For both models, prior uses most of the components to produce a complex distribution. Also, notice that the components learned different non-uniform weights.

## 6.3 Generated Images

Here, we visualize the correspondence of modes and generated images by a procedure that we call *mode hopping*. We start by randomly sampling a latent code and producing the first image. After that, we randomly select five dimensions and sample them conditioned on the remaining dimensions. We

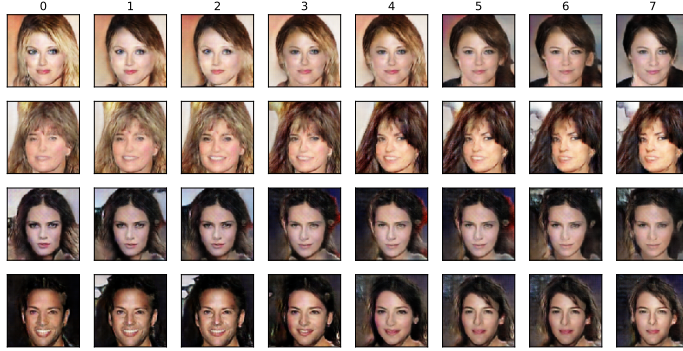

Figure 5: Mode hopping in WGAN-GP-TRIP. We start with a random sample from the prior and conditionally sample five random dimensions on each iteration. Each line shows a single trajectory.

repeat this procedure multiple times and obtain a sequence of sampled images shown in Figure 5. With these results, we see that similar images are localized in the learned prior space, and changes in a few dimensions change only a few fine-grained features.

## 6.4 Generated Conditional Images

In this experiment, we generate images given a subset of attributes to estimate the diversity of generated images. For example, if we specify 'Young man,' we would expect different images to have different hair colors, presence and absence of glasses or hat. Generated images shown in Figure 3 indicate that the model learned to produce diverse images with multiple varying attributes.

Table 3: Generated images with VAE-TRIP for different attributes.

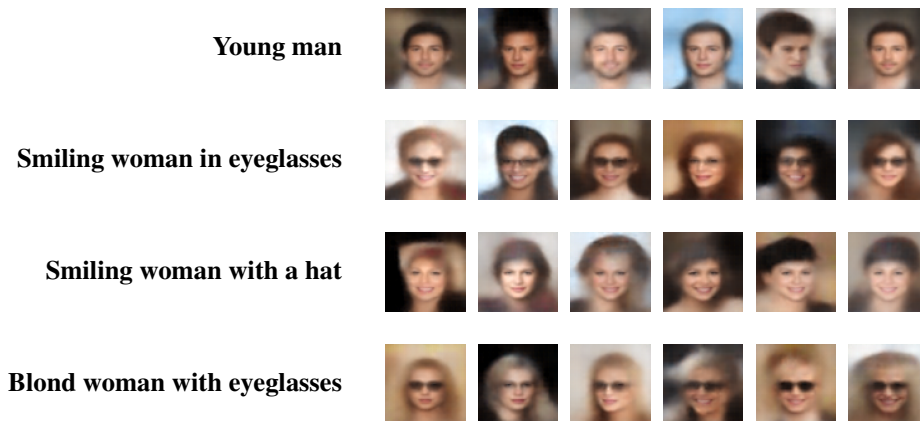

## 7 Discussion

We designed our prior utilizing Tensor Ring decomposition due to its higher representation capacity compared to other decompositions. For example, a Tensor Ring with core size $m$ has the same capacity as a Tensor-Train with core size $m^2$ [35]. Although the prior contains an exponential number of modes, in our experiments, its learnable parameters accounted for less than $10\%$ of total weights, which did not cause overfitting. The results can be improved by increasing the core size $m$; however, the computational complexity has a cubic growth with the core size. We also implemented a conditional GAN but found the REINFORCE-based training of this model very unstable. Further research with variance reduction techniques might improve this approach.

## 8 Acknowledgements

Image generation for Section 6.3 was supported by the Russian Science Foundation grant no. 17-71-20072.

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
