[Supplementary Material · A Prior of a Googol Gaussians_Supplementary.pdf]

# A Prior of a Googol Gaussians: a Tensor Ring Induced Prior for Generative Models (Supplementary)

Maksim Kuznetsov[1,*]    Daniil Polykovskiy[1,*]    Dmitry Vetrov[2]    Alexander Zhebrak[1]

[1]Insilico Medicine   [2]National Research University Higher School of Economics
{kuznetsov,daniil,zhebrak}@insilico.com   vetrovd@yandex.ru

## 1   Derivations for one-dimensional conditional distributions

In the paper, we stated that one-dimensional conditional distributions are Gaussian Mixture Models with the same means and variances as priors, but with different weights $p_\psi(s_k \mid z_{1:k-1})$. With Tensor Ring decomposition, we can efficiently compute those weights (we denote $\prod_{j=k+1}^{d} \widetilde{Q}_j$ as $\widetilde{Q}_{k+1:d}$):

$$
\begin{aligned}
p_\psi(s_k \mid z_{1:k-1}) &\propto p_\psi(s_k, z_{1:k-1}) \\
&= \sum_{s_{1:k-1}} p_\psi(s_{1:k-1}, s_k, z_{1:k-1}) \\
&= \sum_{s_{1:k-1}} p_\psi(s_{1:k}) p_\psi(z_{1:k-1} \mid s_{1:k-1}) \\
&\propto \sum_{s_{1:k-1}} \mathrm{Tr}\left( \prod_{j=1}^{k-1} Q_j[s_j] Q_k[s_k] \widetilde{Q}_{k+1:d} \right) \prod_{j=1}^{k-1} p_\psi(z_j \mid s_j) \\
&= \mathrm{Tr}\left( \sum_{s_{1:k-1}} \prod_{j=1}^{k-1} \left[ Q_j[s_j] p_\psi(z_j \mid s_j) \right] \cdot Q_k[s_k] \widetilde{Q}_{k+1:d} \right) \\
&= \mathrm{Tr}\left( \prod_{j=1}^{k-1} \left( \sum_{s_j} Q_j[s_j] p_\psi(z_j \mid s_j) \right) \cdot Q_k[s_k] \widetilde{Q}_{k+1:d} \right)
\end{aligned}
\tag{1}
$$

## 2   Calculation of marginal probabilities in Tensor Ring

In Algorithm 1 we show how to compute marginal probabilities for a distribution parameterized in Tensor Ring format. Note that we compute a normalizing constant on-the-fly.

## 3   Model architecture

We manually tuned the hyperparameters: first we selected the best encoder-decoder architecture for a Gaussian prior and then tuned TRIP parameters for a fixed architecture. For models from a GAN family, we used a deconvolutional generator with kernel size $5 \times 5$ and ReLU activations. The number of channels in layers was $[512, 256, 128, 64, 3]$. For the discriminator, we used the symmetric convolutional architecture with a LeakyReLU. We trained a model using Adam [1]

---

---
**Algorithm 1** Calculation of marginal probabilities in Tensor Ring
---
  **Input:** A set $M$ of variable indices, values of these variables $r_i$ for $i \in M$
  **Output:** Joint probability $\log p(r_M)$, where $r_M = \{r_i \ \forall i \in M\}$
  Initialize $Q_{\text{buff}} = I \in \mathbb{R}^{m_1 \times m_1}$, $Q_{\text{norm}} = I \in \mathbb{R}^{m_1 \times m_1}$
  **for** $j = 1$ **to** $d$ **do**
    **if** $j$ is marginalized out ($j \notin M$) **then**
      $Q_{\text{buff}} = Q_{\text{buff}} \cdot \left( \sum_{s_j=0}^{N_j-1} Q_j[s_j] \right)$
    **else**
      $Q_{\text{buff}} = Q_{\text{buff}} \cdot Q_j[r_j]$
    **end if**
    $Q_{\text{norm}} = Q_{\text{norm}} \cdot \left( \sum_{s_j=0}^{N_j-1} Q_j[s_j] \right)$
  **end for**
  $\log p = \log \text{Tr}\,(Q_{\text{buff}}) - \log \text{Tr}\,(Q_{\text{norm}})$
---

optimizer with a learning rate of $0.0001$ for $100000$ iterations with a batch size $128$. We used a schedule of $4$ discriminator updates per one generator update. A TRIP prior was $128$-dimensional with $10$ Gaussians per dimension and core size $m_k = 40$ (sizes of matrices $Q_k[s_k]$). For a baseline Gaussian Mixture Model (GMM) prior we used $128 \cdot 10 = 1280$ Gaussians. We conducted all the experiments on Tesla K80.

For VAE models, we used a convolutional encoder and a deconvolutional decoder with a kernel size $5 \times 5$, and the number of channels $[3, 64, 128, 256, 512]$ for the encoder, and a symmetrical architecture for the decoder. We used LeakyReLU for the encoder and ReLU for the decoder. We trained the model for $80,000$ weight updates with batch size $128$. The latent dimension was $100$ for all VAE-based models. For TRIP we used $10$ Gaussians per dimension and a Tensor Ring with core size $m_k = 20$. For a GMM prior we used $1000$ Gaussians.

For conditional generation with TRIP, the architecture was the same as for unconditional generation. For CVAE we parameterized a posterior model $p_\psi(z \mid y)$ as a fully connected network with layer sizes $[2, 128, 100]$ and LeakyReLU activations. For the VAE TELBO baseline model [2], we used a fully connected network for $p_\psi(y \mid z)$ with layer sizes $[100, 64, 64, 2]$ and LeakyReLU activations.

## 4 Implementation details

Implementing the TRIP module is straight-forward and requires two functions. The first function that we use during training computes $\log p_\psi(z_M)$ for an arbitrary subset $M$ of latent dimensions. The second function is used for sampling, and samples from $p_\psi(z)$ with a chain rule, for which calculations are described in Eq 1.

During training we enforce values of cores $Q$ to be non-negative by replacing each element of tensors $Q$ with their absolute values before computation. To make computations more stable, we divide $Q_{\text{buff}}$ and $Q_{\text{norm}}$ by the $\|Q_{\text{buff}}\|$ at each iteration when computing $\log p_\psi(z)$.

Table 1: Impact of core size $m_k$ (CIFAR-10 and CelebA)

| $m_k$ | CIFAR-10 | | | CelebA | | |
|---|---|---|---|---|---|---|
| | ELBO | Reconstruction | KL | ELBO | Reconstruction | KL |
| 1 | -89.5 | 60.5 | 29.0 | -243.40 | 177.63 | 65.76 |
| 5 | -89.3 | **60.2** | 29.1 | -231.57 | 166.89 | 64.67 |
| 10 | -89.3 | 60.4 | **28.9** | -223.59 | **156.99** | 66.60 |
| 20 | **-89.1** | **60.2** | **28.9** | **-215.62** | 158.95 | **56.67** |

# 5 Impact of core size

In Table 1 we compared the performance of VAE-TRIP model with different core sizes $m_k$ on CIFAR-10 and CelebA datasets. Note that for $m_k = 1$, TRIP is factorized over dimensions, where each dimension is a 1D Gaussian Mixture Model. Notice that models with higher core sizes perform better as the prior becomes more complex. In Table 2 we show computational complexity and memory usage of TRIP model to illustrate a tradeof between quality and computational complexity of the model.

Table 2: Time and memory consumption of operations with prior (per batch). $m_k$ is a core size, latent space dimension $d = 100$, number of Gaussians per dimension $N = 10$, batch size $b = 128$. Other parameters are the same as used in the paper. We performed the experiments on Tesla K80. MS stands for milliseconds, MB stands for megabytes. Results averaged over 10 runs; Reported mean $\pm$ std.

| $m_k$ | LOG-LIKELIHOOD, MS | SAMPLING, MS | MEMORY, MB |
|---|---|---|---|
| $\mathcal{O}$-NOTATION | $O(b \cdot d \cdot (m_k^3 + m_k^2 N + N))$ | | $O(d \cdot (m_k^2 + N))$ |
| 1 | $126 \pm 7$ | $201 \pm 21$ | 0.023 |
| 10 | $137 \pm 4$ | $232 \pm 13$ | 0.77 |
| 20 | $193 \pm 15$ | $312 \pm 18$ | 3.1 |
| 50 | $200 \pm 20$ | $360 \pm 17$ | 19.5 |
| 100 | $308 \pm 12$ | $882 \pm 15$ | 78.1 |

Table 3: Condition satisfaction (accuracy) for conditional generative models with different rates of missing attributes in the training set.

| MODEL | % MISSING | | |
|---|---|---|---|
| | 0% | 90% | 99% |
| CVAE [3] | 86.69 | 85.31 | 84.61 |
| VAE TELBO [2] | 82.80 | 74.87 | 73.92 |
| JMVAE [4] | 81.87 | 80.65 | 73.68 |
| VAE-TRIP (OURS) | **88.7** | **87.08** | **84.89** |

## 5.1 Conditional Generation

For the conditional generation, we used images of size $64 \times 64$. We study the model performance for different rates of missing attributes ($0\%$, $90\%$, $99\%$). For each model, we generated 30,000 images for randomly sampled complete sets of attributes from the test set. We trained a predictive convolutional neural network on a validation set to predict the attributes with $92.3\%$ accuracy and predicted the attributes of generated images. We report the condition matching accuracy—when requested attributes matched the actual attributes. We trained all models except for CVAE [3] directly on data with missing attributes. For CVAE, we imputed missing values with a predictive model. For the missing rate of $90\%$, the predictive test accuracy was $90\%$, and for $99\%$—$87\%$. In the results shown in Table 3, we see that the VAE-TRIP model outperforms other baselines.

Table 4: Preliminary results on combining TRIP and normalizing flows to form a prior; Number of parameters of model components

| | $\mathcal{N}(0,1)$ | GMM | TRIP | COMBINATION WITH FLOW | | |
|---|---|---|---|---|---|---|
| | | | | $\mathcal{N}(0,I)$ | GMM | TRIP |
| PARAMETERS (MODEL) | 11.4M | 11.1M | 10.7M | 11.3M | 10.7M | 10.4M |
| PARAMETERS (PRIOR) | 0 | 0.2M | 0.6M | 0.3M | 0.5M | 0.7M |
| PARAMETERS (TOTAL) | 11.4M | 11.3M | 11.1M | 11.5M | 11.2M | 11.1M |
| ELBO | -192.6 | -190.05 | -189.1 | -185.3 | -186.0 | **-184.7** |

## 5.2 Additional experiments for VAE

In Table 4 we compare VAE model with Gaussian, GMM and TRIP priors with a comparable number of parameters. We also provide preliminary results on combining normalizing flows with a TRIP prior.