[Reviews · NeurIPS 2019]

Reviewer 1



I have read the author response and other reviews and decided to keep my original score of 7. Summary: The paper proposes a family of priors for GANs and VAEs. These priors are mixtures of Gaussians with a large number of components but which can be represented using few number of learnable parameters using tensor ring decomposition. This family of priors enable efficient marginalization and conditioning. The method is applicable to both discrete and continuous latent variables. The method is extended to conditional generative modeling; in particular missing values in the conditioning variable can be marginalized out. Experiments are conducted on CelebA and Cifar10. Originality: The proposed method is novel to my knowledge. Clarity and Quality: The paper is very well written and easy to follow. The experiments are somewhat satisfying. I would have liked to see comparison to works using richer priors. For example comparison to the VampPrior [1] for the VAE experiment would be useful. Furthermore, it is not clear whether the TRIP outperforms the GMM baseline solely because it has higher capacity. For example in the appendix it is mentioned that for the GMM the number of components used is 1000; I was expecting 128*10 number of components (128 dimensional latents with 10 gaussians for each dimension). See section 3 of the supplement. Significance: For the VAE, I would deem this work significant if it was shown that this has the possibility to also help with latent variable collapse. For the GAN I would deem this work less significant as it relies on REINFORCE which is somewhat problematic due to high variance (this is rightfully acknowledged in the paper). Questions and Minor Comments: (1) What happens when you use this approach to form the variational distribution in the VAE? (2) line 100: it is "log marginal likelihood" not "marginal log-likelihood" (3) For the GAN did you also use multiple samples from the prior as a GAN baseline? (4) Why not use 1280=128*10 for the GMM baseline in the gan model? That would be more fair to the baseline. (5) How do you select the core size m_k? [1] VAE with a VampPrior. Tomczak and Max Welling, 2018.

Reviewer 2



Thank you to the authors for performing these experiments and addressing the concerns raised by the reviewers. I am pleased to see the performance of TRIP in the context of flows as well. I recommend that this paper be accepted. == The authors present TRIP (Tensor Ring Induced Prior), a parametric family of distributions. These distributions are parameterized as a tensor ring decomposition (Zhao et al. 2016) by d "cores," which define a distribution over d discrete variables. A continuous distribution over R^n can be obtained by placing one Gaussian distribution for each value of the discrete variables, which corresponds to a mixture of a very large number of Gaussians (10^100 Gaussians in this paper). The authors then demonstrate the effectiveness of this parameterization as a learnable prior for VAEs and GANs. The authors justify this approach because the inherent multimodality of this parameterization may better suit the multimodal nature of natural images. The authors cite half-present glasses in the case of GANs trained on CelebA as a disadvantage of unimodal priors. Originality: This work is builds on a wide body of work on learned priors. This approach seems novel as far as I'm aware, although I'm not familiar with the related work on tensor decompositions. Quality: The authors carefully motivate, define, and experimentally test this approach in a wide variety of settings. One concern I have about the experimental setup is that the authors compare TRIP to a N(0, I) prior and a GMM prior. However these seem like unfair comparisons because TRIP has many more parameters than N(0, I) and GMM. It may be fairer to compare to a decoder with the same number of parameters as a TRIP-based decoder would have. I would also like to have gotten a better sense of how much slower a TRIP prior is to train compared to the standard approaches. Clarity: I found this paper to be well-written and easy to follow. The logic flows well from section to section. I very much appreciated the visualizations, especially Figure 1, 4, and 5. Significance: TRIP seems like a practical algorithm that can be used as a prior for VAEs and GANs, or more generally whenever a mixture of a large number of Gaussians is desired.

Reviewer 3



# Overall This paper introduces a complex prior (TRIP) for deep generative models. TRIP has tractable marginal and conditional distributions and can produce an exponential number of mixtures of Gaussian with a small number of parameters. Overall, the paper is well written, the proposed technique is elegant and the motivation is clear. The main weakness is the experiment. # Weaknesses - Some important related works are discussed in Sec.5 but not compared directly in the experiments. What is gained by TRIP vs autoregressive priors [12,13] or flow-based priors [15]? There are no quantitative comparisons between training the generative models with TRIP and with other advanced parametrized priors. - What is the computational cost of TRIP? Since TRIP introduces additional parameters for the prior and brings extra computation, it is worth knowing that how much it slows down the training.

[Author Response · NeurIPS 2019]

Dear reviewers, thank you for a thorough review of our paper. We provide a point-by-point response to each reviewer below.

**Reviewer 1**

1. Using TRIP as a variational distribution is an interesting direction of further research, although we will not be able to apply a reparameterization trick for a TRIP proposal in a way it is used in Gaussian proposals. We will have to use REINFORCE, which may lead to a high gradient variance and, hence, unstable learning.

2. Corrected.

3. For GAN-GMM and GAN-TRIP, we used baselines to reduce REINFORCE's gradient variance (see Eq. 10). A prior of GAN-$\mathcal{N}(0, I)$ is not trainable and hence does not require a baseline. We will add clarification about using baselines to train GAN-GMM to the paper.

4. We thank the reviewer for suggesting to use $128 * 10$ components in the GMM baseline. 1000 components stated in the paper is a typo, the actual number of components was indeed 1280, see the source code file `train_gans.py` from supplementary materials, line 103. We will fix the typo in the paper.

5. We chose the core size to balance computational complexity and empirical performance (see Table 1 below). For $m_k = 20$ the model converged after around one day of training, while for $m_k = 50$ training takes around a week, since it requires more epochs to converge.

Table 1: Time and memory consumption of operations with prior (per batch). $m_k$ is a core size, latent space dimension $d = 100$, number of Gaussians per dimension $N = 10$, batch size $b = 128$. Other parameters are the same as used in the paper. We performed the experiments on Tesla K80.

| $m_k$ | $\mathcal{O}$-NOTATION | 1 | 10 | 20 | 50 | 100 |
|---|---|---|---|---|---|---|
| LOG-LIKELIHOOD, MS | $O(b \cdot d \cdot (m_k^3 + m_k^2 N + N))$ | $126 \pm 7$ | $137 \pm 4$ | $193 \pm 15$ | $200 \pm 20$ | $308 \pm 12$ |
| SAMPLING, MS | | $201 \pm 21$ | $232 \pm 13$ | $312 \pm 18$ | $360 \pm 17$ | $882 \pm 15$ |
| MEMORY, MB | $O(d \cdot (m_k^2 + N))$ | 0.023 | 0.77 | 3.1 | 19.5 | 78.1 |

The reviewer also asked to test the TRIP model for a posterior collapse. For a multimodal prior, a posterior collapse is indeed unlikely, since we cannot approximate a multimodal distribution with a single mode; the only failure mode is when prior collapses to a unimodal distribution along some axis. For our VAE-TRIP model, the number of active units (AU) was $100/100$. We will also add an experiment on MNIST and StackedMINST to a camera-ready version.

**Reviewer 2**

1. The reviewer suggested benchmarking the models with TRIP, GMM, and Gaussian priors with the same number of parameters. We present the result of this experiment in Table 2 below, supporting the conclusions we got from the original experiment.

Table 2: VAEs with different priors and a comparable number of parameters

| | $\mathcal{N}(0, 1)$ | GMM | TRIP | $\mathcal{N}(0, I)$-FLOW | GMM-FLOW | TRIP-FLOW |
|---|---|---|---|---|---|---|
| PARAMETERS (MODEL) | 11,4M | 11,1M | 10,7M | 11.3M | 10.7M | 10.4M |
| PARAMETERS (PRIOR) | 0 | 0,2M | 0,6M | 0.3M | 0.5M | 0.7M |
| PARAMETERS (TOTAL) | 11,4M | 11,3M | 11,1M | 11.5M | 11.2M | 11.1M |
| ELBO | -192.6 | -190.05 | -189.1 | -185.3 | -186.0 | -184.7 |

**Reviewer 3**

1. The proposed TRIP model has many useful properties such as conditioning on a subset of attributes (Sec. 4)—a property that other priors (including flow-based models) do not have. For a fair comparison, we incorporated TRIP as an initial distribution of a flow-based RealNVP prior and show in Table 2 that such model outperforms a standard RealNVP prior. We will add a section on incorporating neural priors to the updated paper, including VAMP and IAF priors.

2. The computational costs of TRIP depend on the number of dimensions $d$ and core size $m_k$ (usually constant for all $k$). We report asymptotic complexities, time, and memory measurements in Table 1, showing that TRIP is practical for moderate core sizes.

[Meta-Review · NeurIPS 2019]

The paper introduces a novel way of parameterizing a mixture of Gaussians with exponentially many modes by using the Tensor Train decomposition to capture the dependence between the mixing variables of the per-dimension 1D Gaussian mixtures. The resulting distribution, which supports efficient marginalization and conditioning, is then used as a prior in VAEs and GANs. The reviewers agreed that the idea is novel and interesting and the paper is well written. The authors have addressed in the rebuttal some of the reviewer concerns about the mismatch in the number of parameters in the proposed prior and the baseline priors. The main remaining weakness of the paper is the lack baselines with strong priors (e.g. autoregressive or expressive flows), though the flow-based baseline provided in the rebuttal is a reasonable first step in that direction.